# Development of a Real-Time Continuous Measurement System for Tree Radial Direction

**Qianjia Zhang [1,2], Yuanjing Sun [1,2], Xinyu Zheng [1,2], Shusheng Zhang [3,\*] and Luming Fang [1,2,\*]**

1   College of Mathematics and Computer Science, Zhejiang A&F University, Hangzhou 311300, China; 2020611011064@stu.zafu.edu.cn (Q.Z.); 202061101104@stu.zafu.edu.cn (Y.S.); zhengxinyu@zafu.edu.cn (X.Z.)
2   Zhejiang Provincial Key Laboratory of Forestry Intelligent Monitoring and Information Technology, Hangzhou 311300, China
3   Management Center of National Natural Conservation Area in Zhejiang Wuyanlin, Taishun 325500, China
\*   Correspondence: zhangshusheng323@163.com (S.Z.); flming@zafu.edu.cn (L.F.);
    Tel.: +86-181-5776-8179 (S.Z.); +86-189-6815-6768 (L.F.)

**Abstract:** Tree diameter at breast height (DBH) is the most fundamental factor in modelling tree growth, but current DBH measurement instruments mainly focus on instantaneous acquisition, making it difficult to measure tree growth continuously and accurately. In this study, we propose a wireless sensing network that can transmit data in signal-free environments, and combine sensor and computer technologies to develop a real-time continuous measurement system for tree radials, which has the advantages of working in real-time, being low-cost and stable, and enabling high-precision. It can be applied to the DBH measurement of trees in the range of 50 mm–380 mm, with a measurement accuracy of 0.001 mm. Additionally, whole-point sampling, conducted 24 h per day, integrates DBH data measurement, transmission, storage and visualization analysis. After measuring in the field for a year, it initially reveals the change in DBH within the test area within that year. This study provides a scientific basis for researching the microscopic growth pattern of trees and establishing a tree growth model, which will be further optimised and improved in terms of appearance structure, communication and power supply in the future.

**Keywords:** displacement sensor; wireless transmission; data management; pattern of change in breast diameter

## 1. Introduction

Changes in tree diameter at breast height (DBH) indicate the physiological state of trees and their response to external factors [1]. Changes in tree DBH contain information about tree growth and water use [2], and can be used to explore the interaction between trees and climatic indicators. With the development of electronic information and computer technology, a large number of new measurement methods, which are mainly categorized into two main types including the instantaneous measurement and the continuous measurement, were derived, and the instantaneous measurement is mainly categorized into three main types: remote sensing measurement, embedded equipment measurement and photogrammetry. For remote sensing measurement, Liang [3], Huang [4], etc. mainly used remote sensing data to generate multiple feature bands based on multi-information fusion, and then combined them with the actual measurement factors of the forest field to obtain the measurement data, which are then applied in forestry measurement; or utilized terrestrial LiDAR (LiDAR) [5–9], using the multi-single-shot (MSS) method, for trunk scanning and point cloud data extraction, and then fit the circle and point cloud data to obtain the value of breast diameter. For laser embedded devices, Linhao Sun [10,11], Fangxing Yuan [12,13], Shangyang Li [14], etc. mainly used microcontrollers, sensors, integrated circuits in combination with electronic information and computer technology to produce corresponding devices in order to realize the convenient, timely and fast measurement

of tree diameter at breast height, whereas for photogrammetry in the field, Close Range Photography (CRP) [15–17] and smartphones with TOF cameras [18–20] were mainly used to extract tree diameter at DBH by scanning the tree trunk in all directions, thus simulating the human eye to measure the spatial position and size of the object.

The instantaneous measurement methods and devices can solve the shortcomings of time-consuming and labour-intensive traditional measurement methods, but fail to satisfy the real-time mastery of tree growing stock volume data. Forestry researchers have begun to explore the continuous measurement of tree diameter, but the overall research in this area is relatively small. The DC series of DBH measuring equipment produced by Ecomatik in Germany is suitable for trees with a DBH of 5 cm or more, but the equipment can only measure the diameter change of the trunk in a certain direction [21]. Meanwhile, the Czech company READING ENVIRO developed the DRL26C series of continuous diameter at breast height (DBH) measuring devices, which are highly accurate and equipped with an integrated angular displacement sensor with a resolution of 0.001 mm, and are suitable for stumps with DBH of more than 8 cm. However, the data of the above two continuous measurement devices of breast diameter are saved to the local device and still need to be collected manually.

The various types of chest diameter measurement equipment and methods are summarised in Table 1.

**Table 1.** Summary of equipment and methods for measuring chest diameter.

|  | Instant Measurement | | | Continuous Measurement |
|---|---|---|---|---|
|  | Remote Sensing Measurements | Embedded Device Measurement | Photogrammetric |  |
| Measurement characteristics | DBH values can be estimated using remote sensing images, making data acquisition easy and fast. | Based on a variety of electronic sensors, the tree is regarded as a circle-like shape for measurement to obtain the DBH value, which is easy to operate and portable. | Using special filming equipment to simulate the human eye to measure the spatial position of objects, the DBH of trees is extracted, which is convenient and easy to operate. | Sensor-based tree diameter at breast height (DBH) can be collected over a long period of time, providing DBH values at a high-tempora resolution. |
| Service to be improved | More expensive equipment required, susceptible to environmental influences, longer data processing time, only single data acquisition | It needs to be saved locally on the device and manually uploaded to the server, and only single data can be obtained. | Some of the device cameras are more expensive and have to wait for the algorithm to process the data to obtain only a single shot. | Data is saved locally to the device and manual data collection is required. |

Aiming at the problems of DBH measurement devices mentioned above, this paper focuses on the realisation of continuous measurement of tree DBH, and develops a tree radial real-time continuous measurement system, which mainly consists of three parts, namely data acquisition device, wireless sensor network and data management system, and develops a wireless sensor network that can transmit the data under the no-signal condition. It integrates the functions of tree DBH continuous measurement, wireless transmission and management analysis. The developed device has the advantages of high precision, low-power consumption, low cost, easy installation, real-time transmission and strong anti-interference, etc. The developed software has the advantages of easy operation, high reliability and strong interactivity.

## 2. Materials and Methods

### 2.1. Mechanical and Circuit Structural Design

The DBH continuous measurement device mainly consists of a data acquisition device and a coordinating base station, both of which are made of IP65 ABS material [22], with good dustproof and waterproof properties and high-temperature resistance, to meet the needs of continuous measurement in the field under complex environments. The data acquisition device integrates the DBH data acquisition and data transmission functions, and its main mechanical structure is shown in Figure 1a, while the coordinating base station integrates data aggregation and data transmission functions, and its main mechanical structure is shown in Figure 2a. The data acquisition device circuit is mainly composed of the main control module, power supply module, data acquisition module and communication timing module, etc., in order to minimize the power consumption of the equipment; the main control chip selects the STC15F series, as shown in Figure 1b; the coordinated base station circuit is mainly composed of the main control module, power supply module, data acquisition module and other modules in order to improve the stability of the data and real-time; and the main control chip selects the STC15W4K series, as shown in Figure 2b. The physical figure is shown in Figure 3.

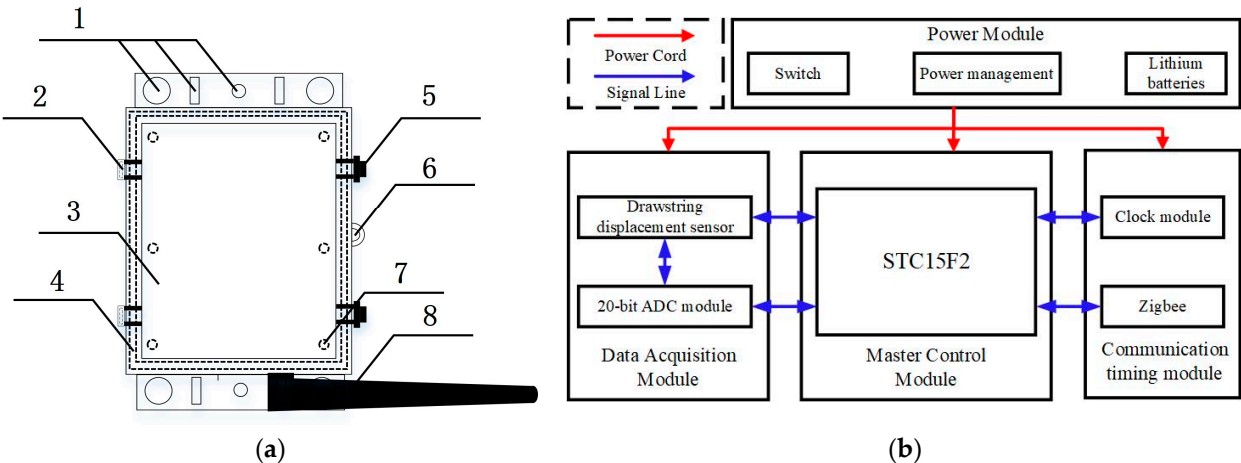

**(a)**            **(b)**

**Figure 1.** Mechanical and Circuit Architecture Design of Data Acquisition Devices. (**a**) Mechanical structure of the data acquisition device: 1. Fixed holes; 2. Hinge spindle; 3. Control box; 4. Silicone sealing strip; 5. Hinge snap; 6. Lock hole; 7. Self-tapping screws; 8. Glue stick antenna. (**b**) Data Acquisition Unit Circuit Architecture.

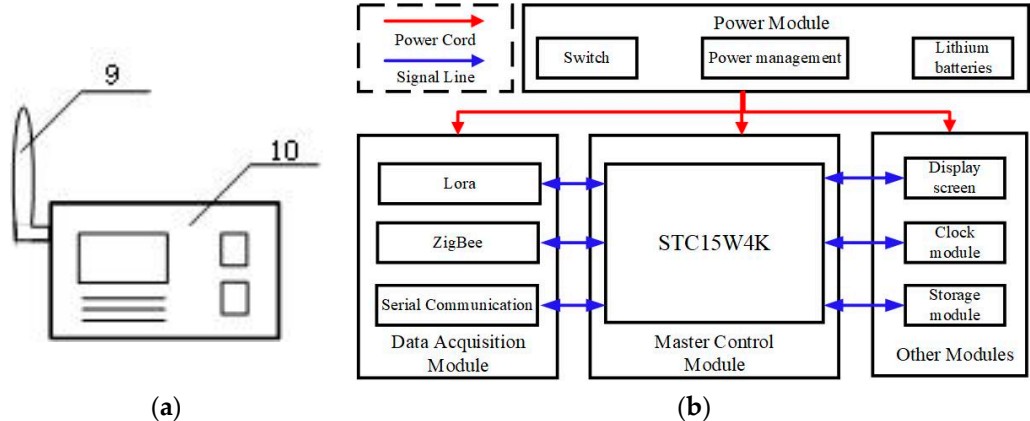

**(a)**            **(b)**

**Figure 2.** Coordination of base station mechanical and circuit architecture design. (**a**) Coordinated base station mechanical structure: 9. Glue stick antenna; 10. Main body box. (**b**) Coordinated Base Station Circuit Architecture.

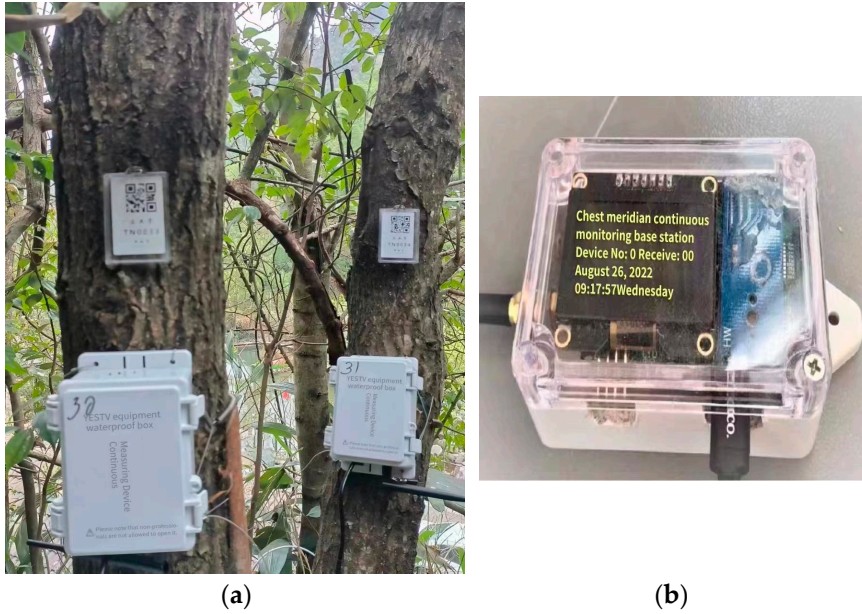

(**a**)　　　　　　　　　　　　　　　　　　(**b**)

**Figure 3.** Physical drawing of continuous measurement device. (**a**) Physical drawing of the data acquisition device. (**b**) Physical drawing of the coordinated base station.

## 2.2. Software Design

In order to realize better uploading, storing and visualizing the data, a set of data management software was developed, which consisted of PC upper computer and a data management platform. The main function of PC upper computer is to serve as a data relay station between the front-end instantaneous sensing device and the database, which will coordinate the data sent by the base station, and through the UART protocol, the data will be received, displayed and uploaded to the database. The upper computer interface is shown in Figure 4, and the main components are: serial port settings, device information, parameter recording and data display. The main functions of the data management platform are visual display, simple analysis, and data query and management, etc. The interface is shown in Figure 5, and its main components encompassed data query, data display, data analysis, etc.

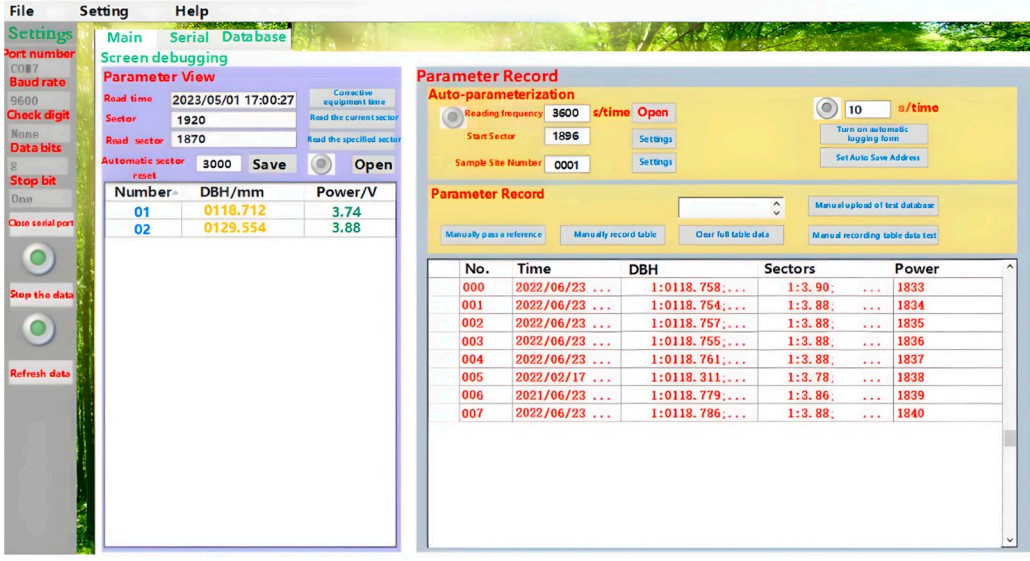

**Figure 4.** Host computer interface.

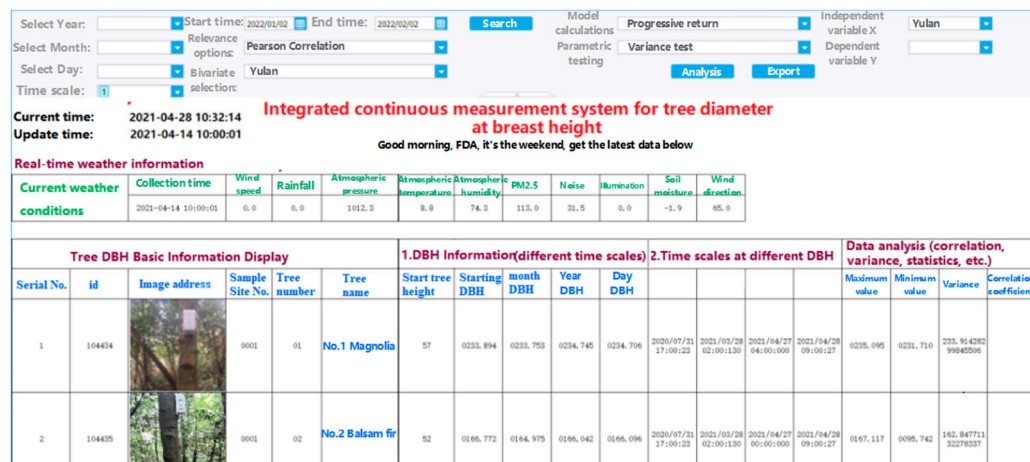

**Figure 5.** Data management platform interface.

*2.3. System Process Design*

The tree DBH continuous measurement system from the bottom up is mainly divided into perception layer, network layer, application layer, and perception layer for the tree DBH collection device, where the tree radial DBH change data are measured and sent to the network layer. The network layer coordinates the base station and host computer, collecting and displaying data, and the host computer is then uploaded to the application layer. The application layer is responsible for the cloud servers, data management platforms, and application software, however, it is ultimately responsible for data management, visualization and displaying data queries, etc. The system flowchart is shown in Figure 6.

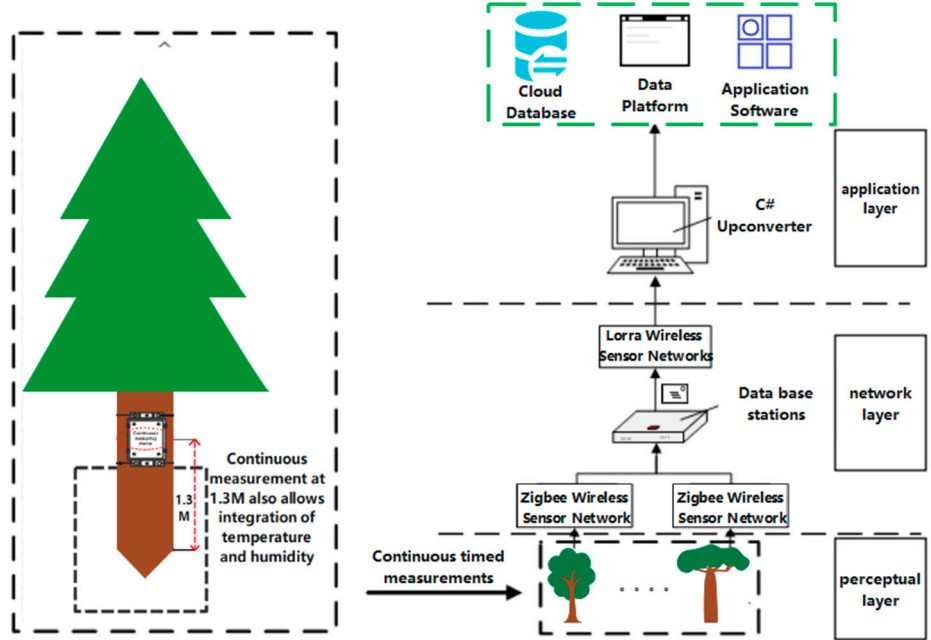

**Figure 6.** System flow chart.

*2.4. DBH Measurement Principle*

The tree DBH measurement was mainly achieved by the pull rope displacement sensor, which inside is equipped with an absolute photoelectric encoder. Through the photoelectric detection element, it is able to determine the position of the engraved line on the code disc. Additionally, the conversion circuit can output the corresponding square wave [23] in which the schematic diagram for this is shown in Figure 7, and then use the analogue-to-digital converter module to obtain the displacement amount *S*.

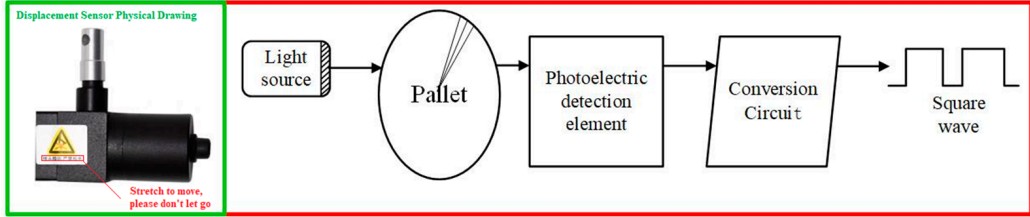

**Figure 7.** Principle of Displacement Measurement for Pull Cord Sensors.

The tree DBH measurement was used for tree trunk approximation as a cylinder, utilising the pull rope displacement sensor, which is surrounded horizontally. At this moment, the state of the pull rope was fixed in order to achieve the asymptotic measurement of the tree DBH. Assuming that the pull rope is around the one-week pull out of the displacement of *S*, and subtracting the error displacement *e*, the known perimeter of the circle is $\pi$. Through Formula (1), the tree measurement can be derived. The size of DBH d is shown in Figure 8.

$$d = \frac{S - e}{\pi} \tag{1}$$

where the DBH of the measured tree, i.e., the circumference $\pi$, means *S* is the displacement Distance, and *e* is the distance from the fixed point to the head of the pull rope.

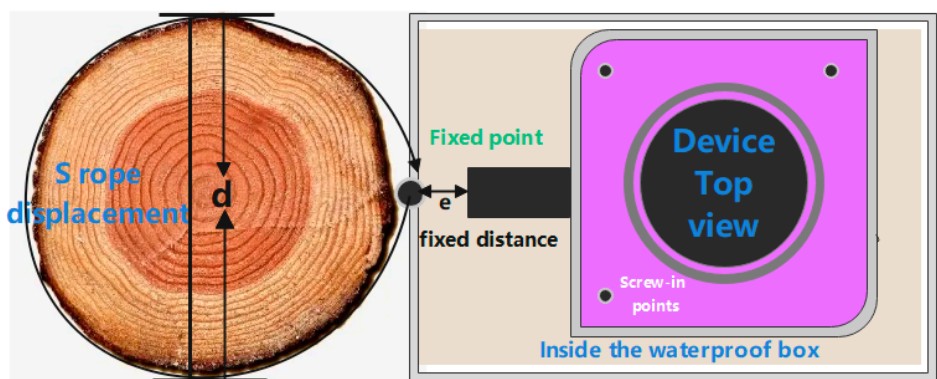

**Figure 8.** DBH Measurement Schematic.

### 2.5. Wireless Sensing Principle

In order to take into account the remote wireless sensing of tree DBH data, a wireless sensing network was established. There are three basic elements in the wireless sensing network: terminal node, coordinating station and user object, in which the terminal ZigBee and sensing module as the terminal node are mainly responsible for collecting and sending tree DBH, power and other related data to the coordinating base station. The coordinator ZigBee and Lora, and other communication modules as the coordinating station, are mainly responsible for receiving the data collected by the terminal node and the control instructions of the user object, and then sending the control instructions and data to the user object. The user object can actively or passively receive the collected data, analyse, then make corresponding decisions or behaviours based on the results after analysis. The principle of infinite sensing is shown in Figure 9.

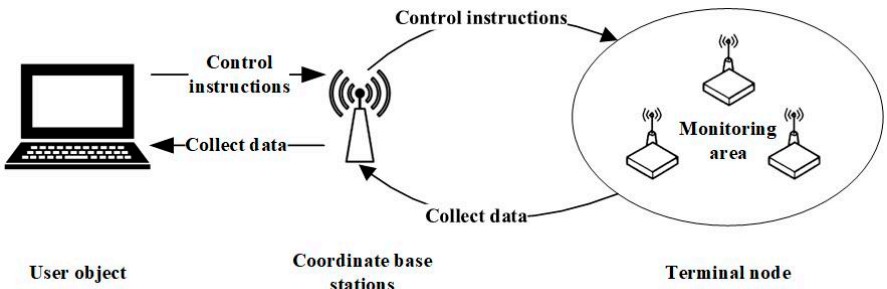

**Figure 9.** Infinite Perception Schematic.

### 3. Tests and Analyses

*3.1. Introduction to the Test Plots*

The test site was located on the campus of Zhejiang Agriculture and Forestry University in Lin'an City, Zhejiang Province (N 30°15′, E 119°43′). At the same fixed sample site, we selected two different trees of similar age, Camphor (also known as zhang shu) and Magnolia (also known as Yulan), to start a one-year (1 August 2021 to 1 August 2022) continuous measurement. At the beginning and end of the measurements, we took manual hand measurements using a tape measure, and three measurements were taken for each tree and averaged, and these data were compared with the data collected by the measurement device. In addition, at each month throughout the measurement period, we also measured the then instantaneous trunk diameter (dbh) using a tape measure and compared it with the data from the measuring device (DBH). According to the measurement results, the errors were kept within less than 0.1 mm, as detailed in Table 2.

**Table 2.** Sample Tree Information Sheet.

| Tree Species | Slope Direction | Slope | Continuous Measuring Device Start DBH | Manual Tape Measure Start dbh | Continuous Measuring Device End DBH | Manual Tape Measure End dbh | Measurement Error |
|---|---|---|---|---|---|---|---|
| Camphor/Zhang shu | North and South Slopes | 30° | 112.242 mm | 112.2 mm | 119.765 mm | 119.8 mm | 0.035 mm |
| Magnolia/Yulan | North and South Slopes | 30° | 126.500 mm | 126.5 mm | 130.362 mm | 130.4 mm | 0.038 mm |

*3.2. Device Measurement Results*

3.2.1. Magnolia and Camphor Annual Changes

Based on the data obtained from the continuous DBH measurement device and the system-generated graphs, Figure 10a shows the changes in DBH of Camphor Tree No. 1 during the period from 1 August 2021 to 1 August 2022. On 29 March 2022, DBH began to rise gradually and continued to rise until the end of the monitoring period, during which time the camphor tree's DBH continued to grow, which is defined as its growth period. The growth pattern of Magnolia 2 was also similar to that of Camphor 1. As shown in Figure 10b, based on the analysis of the above observations, both camphor and magnolia trees in the area are dormant, with the growing period from March to July and the dormant period from August to February. From 1 August 2021 to 21 March 2022, the DBH began to fluctuate slightly, but remained relatively stable in the intermediate period in general. In 21 March 2022, a gradual increase in DBH was noticed, and the procedure continued until the end of the monitoring period.

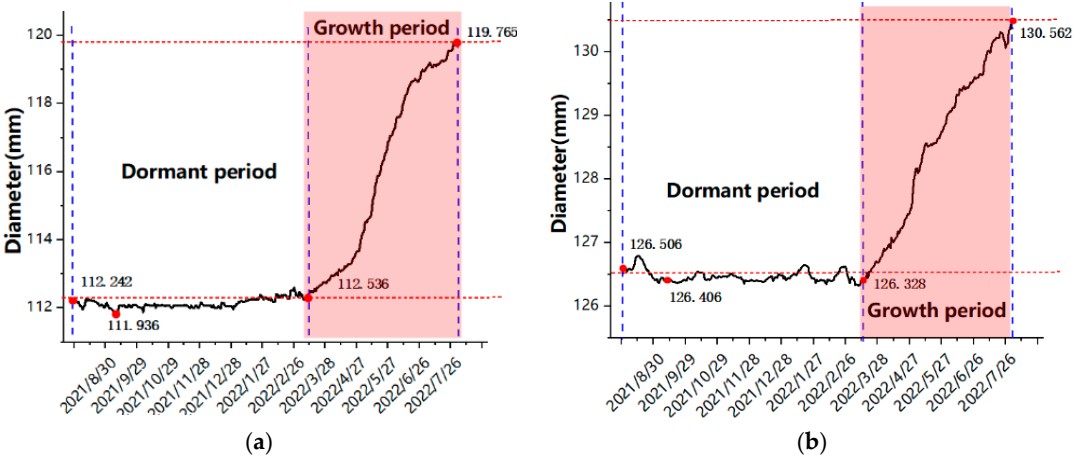

**Figure 10.** Changes in annual diameter at breast height (DBH) values for camphor and magnolia.
(**a**) Changes in one-year diameter at breast height (DBH) values of camphor trees. (**b**) Variation of one
year's diameter at breast height value of magnolia (Magnolia officinalis).

By analysing the above observations, we concluded that the growth and dormant
status of Camphor and Magnolia in the area, with the growth period from March to July and
the dormant period from August to February, meant that the DBH of Camphor increased
by 7.523 mm, while the DBH of Magnolia increased by 4.562 mm.

3.2.2. Magnolia and Camphor Monthly Changes

Based on the data obtained from the continuous DBH measurement system and the
graphs generated by the system, Figure 11 shows the monthly growth of Camphor and
Magnolia from 1 July 2021 to 31 July 2021, where the grey areas represent rainy days. On
3 July 2022, DBH started to increase gradually and continued until 25 July, during which
time DBH continued to increase, and the growth curves showed an S-shape. During this
period, DBH continued to increase and the growth curve showed an S-shaped stepwise
growth. The growth pattern of Yucca 2 was also similar to that of Yucca 1. DBH gradually
levelled off slightly on 25 July 2022, and then increased rapidly on 29 July, which continued
until the 31 July.

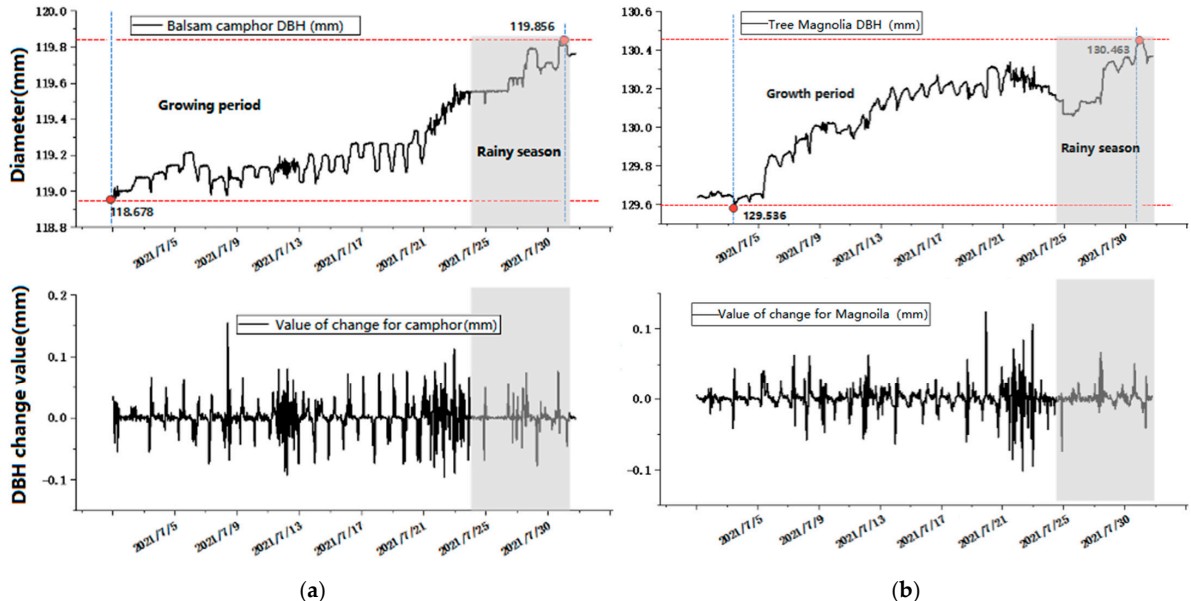

**Figure 11.** Monthly DBH growth of camphor and magnolia trees (grey represents rainy days).
(**a**) Changes in DBH growth in July in camphor. (**b**) Changes in DBH growth of magnolia in July.

By analysing the above observations, we can conclude that the monthly growth of camphor and magnolia in this area showed an S-type stepwise increase, with an average of one day as a cycle, and Camphor cumulatively (1.178 mm) growing faster than Magnolia (0.927 mm) during the month. Additionally, the daily variation of DBH was between −0.05 and 0.15 mm (the value of change is the value at the next moment minus the value at the previous moment, e.g., DBH at 7 o'clock minus DBH at 6 o'clock, which can reflect the speed of growth). Around the same time, the rainy season prolonged the growth cycle of magnolia and camphor, causing the trees to absorb water and swell, and thus, DBH grew rapidly.

### 3.2.3. Magnolia and Camphor Daily Change

Figure 12a shows the single-day variation of DBH in camphor trees in which the DBH value shows an upward trend from 0:00 to 07:00 and 07:00 to 15:00, then a gradual decline from 15:00 to 23:00, and finally, an upward trend again. Figure 12b shows the amount of variation in DBH of magnolia officinalis on a single day. From 0:00 to 06:00 h, DBH values began to show an increasing trend; from 06:00 to 16:00 h, DBH values gradually decreased; from 16:00 to 23:00 h, DBH values began to show an increasing trend again. Changes in DBH values were measured throughout the day for both trees, but the magnitude of the changes was small and the overall trend was upward. Although the DBH values of the two trees changed slightly differently throughout the day, the overall trend was more or less the same, which indicates that the growth patterns of different tree species are slightly different throughout the day. Based on the data analysis, the daily growth pattern of the two trees in this experimental application was roughly the period of DBH growth from 15:00 to 16:00 h to 06:00 to 07:00 h the next day, and the period of DBH slight contraction from 06:00 to 07:00 h to 15:00 to 16:00 h each day. The contraction and expansion of tree DBH during the day is related to water evaporation and uptake due to photosynthesis and transpiration in trees.

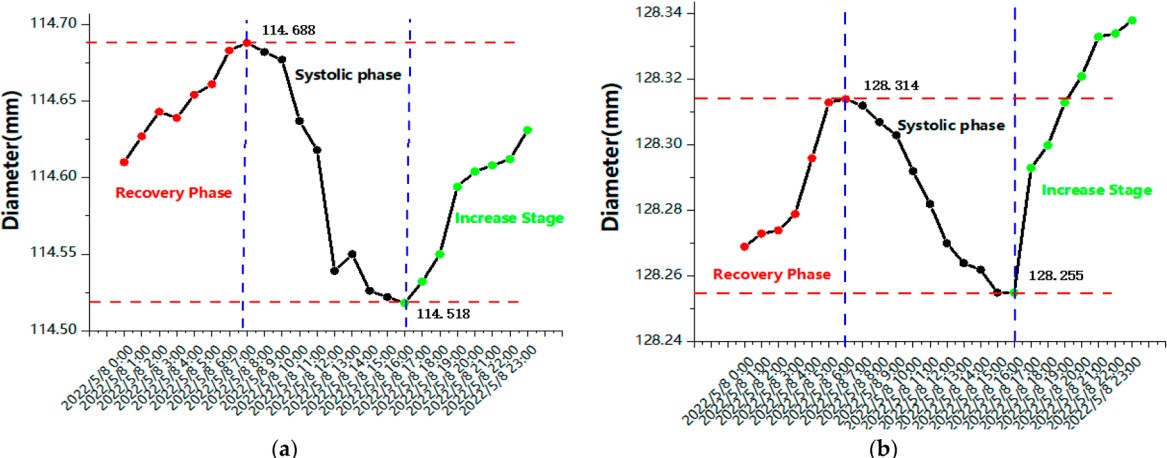

(**a**)  (**b**)

**Figure 12.** Relationship between DBH and atmospheric humidity on a single day (8 May) for camphor and magnolia. (**a**) Changes in DBH growth on a single day at camphor (8 May). (**b**) Variation in DBH growth of magnolia alba on a single day (8 May).

From the above observations and analyses, it can be concluded that the growth of both trees, camphor and magnolia, can be roughly divided into three phases in a day: 24:00–7:00 for the recovery phase, 7:00–15:00 for the contraction phase, and 15:00–23:00 for the rise phase. In the recovery phase, the change was 64–68 μm; in the contraction phase, it was 59–117 μm; and in the ascending phase, it was 85–107 μm with an overall rising S-shaped curve.

## 4. Discussion

For DBH measurement, a large number of scholars have conducted relevant research. For example, Liang et al. used ground-based LIDAR to extract the DBH by fitting the circle and the point cloud data, and obtained the average deviation (BIAS)-0.18-0.76 cm and root mean square error (RMSE) of 0.74–2.41 cm; Huang et al. used LiDAR combined with the circle fitting algorithm to estimate the DBH of standing wood accordingly, and finally determined that the accuracy of DBH was 3.4 cm; Forsman et al. [24] used a multi-camera setup to collect image information in multiple directions by generating a 3D-point cloud in order to extract the DBH of standing timber, and the experimental results of the field measurements showed that the root mean square error (RMSE) of the DBH measurements was 2.8–9.5 cm; Fan [25] et al. used a mobile phone with RGB-D SLAM to extract the point cloud data of tree trunks, and fitted the point cloud data by circles to estimate the DBH, and the RMSE of their experimental results was 0.69 cm with an average deviation of 0.36 cm. The above methods are susceptible to the environment, complicated data processing, and high cost of equipment, which make it difficult to be widely promoted and applied in reality. There are also some researchers who developed embedded devices; for example, Linhao Sun and others, in the traditional perimeter ruler measurement DBH principle based on the absolute encoder, wireless Bluetooth, etc., designed a portable tree DBH measurement equipment, and the device between different species of DBH measurement, resulted in an accuracy of 99.97%; the average error in measurement accuracy was within ±1.7; and the average measurement of each tree consumed 5.3 s. Fangxing Yuan et al. developed an angle-based tree DBH measurement device, called a three-segmented sensor, with a reported deviation of 0.1 cm and an RMSE of 0.45 cm. Embedded devices are suitable for promotion in production practice because of their improved efficiency and accuracy compared to traditional measurement tools, and their lower cost. However, single instantaneous measurements are still not available for the acquisition of dynamic changes in tree DBH.

The continuous measurement of DBH is relatively less studied; for example, Ecomatik of Germany has conducted a more in-depth study on the continuous measurement of tree growth. Its DC series of products offer high and accurate measurements with a resolution of up to 0.2 μm, which is suitable for upright trees with a DBH greater than 5 cm, and is capable of measuring plants without causing damage, and has an easy installation process. However, the device can only measure the DBH change in a certain direction of the tree trunk [26], because the DBH of standing trees will be affected by factors such as light, temperature and humidity, resulting in the phenomenon that the growth change in different radial directions is not the same [27,28], and does not objectively reflect the amount of change in the growth of the tree's DBH. The Czech Republic (EMS, Brno, Czech Republic) developed the DRL26C series of DBH continuous measurement equipment, which integrates an angular displacement sensor with a resolution of 0.001 mm and a high measurement accuracy for stumps with DBH exceeding 8 cm. The product is compact, easy to install, and the product is well encapsulated and waterproof, but the measurement data are saved to a local device and manual data collection is still required.

In this study, a new real-time continuous measurement system for tree radials was developed, combining a drawstring displacement sensor, ZigBee and Lora communication modules, and data management software. It achieves timely and accurate uploading of DBH data to server storage for 24 h, and visual display and analysis. After a year of field measurement verification, the data acquisition device of this system can work stably for a long time under the complex environment in the field, and it is suitable for the dynamic measurement of all kinds of trees with an accuracy of up to 0.001 mm, which is the same as the accuracy of similar products, and the cost is only 1/5 of those, which is about 181.63 CFH. The wireless sensing network can realise the stable transmission of data in the absence of signals, and the data management platform is convenient for users to query, manage and analyse data. Compared with the instant single measurement equipment, this system can dynamically acquire the DBH data of trees. Compared with the continuous

measurement equipment, this system can wirelessly transmit to the server, eliminating the step of manual collection, and supporting the corresponding data management software. This system integrates data acquisition, transmission, storage, visualisation and analysis, and can be used to construct tree growth models and research on tree growth characteristics.

## 5. Conclusions

Based on the drawstring displacement sensor and communication module, we developed a real-time continuous measurement system for tree radial direction and verified its stability and reliability after one year of measurement tests in the field environment. The system can acquire tree diameter at breast height (DBH) data in real time stably, and store, manage and visualise them for analysis. After one year of continuous monitoring, the balsam fir and magnolia trees grow, roughly, simultaneously throughout the year. March to July represents the growing period when DBH rises rapidly, and the rest of the year is the dormant period. Monthly measurements showed an S-shaped sinusoidal increase in DBH growth. The daily data changes showed that the growth of the trees was divided into recovery period (24:00–7:00), contraction period (7:00–15:00) and growth period (15:00–23:00), and the daily DBH variation was about 0.02 mm. This system can reveal the micro-dynamic growth change law of multiple trees, and provide reliable and higher resolution data support for the establishment of the tree growth model. However, this system still has shortcomings that must be improved, and in the next step, we will optimise the structural appearance of the device, and the expandability of the communication method and power supply system.

**Author Contributions:** Conceptualization, L.F. and Q.Z.; Formal analysis, Y.S.; Funding acquisition, S.Z. and Y.S.; Methodology, L.F.; Resources, X.Z. and S.Z.; Writing—original draft, Q.Z. All authors have read and agreed to the published version of the manuscript.

**Funding:** This work was financially supported by the Zhejiang Provincial Key Science and Technology Project (2018C02013); and The National Natural Science Foundation of China, Grant No. 42001354.

**Data Availability Statement:** The data used to support the findings of this study are available from the corresponding author upon request.

**Conflicts of Interest:** The authors declare no conflict of interest.

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
