# Peer review of "Development of a Real-Time Continuous Measurement System for Tree Radial Direction"

_forests, doi:10.3390/f14091876_

Round 1
Reviewer 1 Report (Previous Reviewer 1)
I regard the changes made in the revised version of the manuscript (forests-2594101) as satisfactory and recommend it for publication.
Author Response
尊敬的审稿人 1.
非常感谢您审阅并同意我们的发布。您和第二位审稿人的宝贵意见有助于改进论文。我们很高兴听到您对修订本感到满意,我们将继续努力确保文件的质量和出版准备。
再次感谢您的支持和帮助。
祝你好运!
方陆明和张
倩佳 电子邮件:
202061101164@stu.zafu.edu.cn fluming@126.com

Reviewer 2 Report (New Reviewer)
Dear Authors,
Please see the attached file.

Moderate editing of the English language required
Author Response
Dear reviewer 2.
I hope all is well with you. I would like to thank you very much for your careful review of our paper and for the valuable comments you have provided. I would like to inform you that we have made revisions as per your suggestions one by one and believe that these changes have significantly improved the quality and readiness of the paper. (See attachment: Reviewer comment response letter 2 (see the end for the full text and response letter))
In order for you to review the latest version of the paper, I have attached the revised paper with the answer-by-answer attachments (in order: Title, Keywords, Abstract, Discussion of the paper, Conclusion, and full text embellishments, etc.) to this response. Please feel free to download and review it, and if you have any further comments or feedback, we would greatly appreciate your help again.
We look forward to hearing from you and thank you for your continued interest and support of our research. We will continue to work hard to ensure that the paper is of high quality and ready for publication.
Thank you again for your support and help.
此致敬意。
[审稿人评论回复函2(全文及回复函见结尾)]
祝你好运!
方陆明和张
倩佳 电子邮件:
202061101164@stu.zafu.edu.cn fluming@126.com

Round 2
Reviewer 2 Report (New Reviewer)
There is no comment.
Moderate editing of English language required
This manuscript is a resubmission of an earlier submission. The following is a list of the peer review reports and author responses from that submission.
Round 1
Reviewer 1 Report
Review
The submitted manuscript (MS) deals with the topic of development of a continuous tree diameter at breast height measurement system (Forests-2498558). In my opinion, despite the actual topic the MS does not meet the minimal standards required for an international journal Forests. My concerns regarding the publication of MS are especially due to the unclear goal of the MS, non-standard structure of the article and unsatisfactory quality of the chapter Discussion. In following notes the most important objections will be explained.
1. The goal of the study is not defined. Therefore the reader is not able to identify the key message of the MS.
2. The chapters of the MS are unbalanced regarding their extent. This problem is probably mostly visible in comparison of the chapters Discussion and Conclusion. The MS lacks the standard structure of a scientific paper (Introduction, Methods, Results, Discussion, Conclusions), e.g. a clear specification of chapters „Material and Methods“ and „Results“ is lacking.
3. The chapter Discussion has not the required quality. The authors did not use here a single reference to a scientific paper. Discussion therefore lacks its crucial sense.
4. In the text of the MS many duplicate informations are present (e.g. L 45–47 and L 429–431, L 50–60 and L 417–425, L 350–353 and 461–466).
Regarding my negative statement, I consider the commenting of the detailed point-by-point remarks as irrelevant.
As a conclusion, I think the material presented in this MS is worth publishing in an international journal. Therefore, I encourage the author/s to prepare the improved version of the MS and resubmit it to the journal Forests.
Reviewer 2 Report
The article describes a new device for automatic, continuous measurement of tree breast diameter. The research provides the methods of construction of the device and an experiment with two species.
I recommend the publication after minor revisions below:
Figure 2 needs more resolution.
Figure 3 - include the meaning of "d" and "e" in the legend
line 235 -... Magnolia(note that... add space after the parenthesis